# Noble Metal Complexes of a Bis-Caffeine Containing NHC Ligand

**DOI:** 10.3390/molecules27134316

**Published:** 2022-07-05

**Authors:** Oliver Bysewski, Andreas Winter, Phil Liebing, Ulrich S. Schubert

**Affiliations:** 1Laboratory of Organic and Macromolecular Chemistry (IOMC), Friedrich Schiller University Jena, Humboldtstr. 10, 07743 Jena, Germany; oliver.alexander.bysewski@uni-jena.de (O.B.); andreas.winter@uni-jena.de (A.W.); 2Center for Energy and Environmental Chemistry Jena (CEEC Jena), Friedrich Schiller University Jena, Philosophenweg 7a, 07743 Jena, Germany; 3Institute for Inorganic and Analytical Chemistry (IAAC), Friedrich Schiller University Jena, Humboldtstr. 8, 07743 Jena, Germany; phil.liebing@uni-jena.de

**Keywords:** caffeine, mesoionic carbene, palladium, platinum, NHC

## Abstract

*N*-Heterocyclic carbenes (NHCs) have seen more and more use over the years. The go-to systems that are usually considered are derivatives of benzimidazole or imidazole. Caffeine possesses an imidazole unit and was already utilized as a carbene-type ligand; however, its use within a tridentate bis-NHC system has—to the best of our knowledge—not been reported so far. The synthesis of the ligand is straightforward and metal complexes are readily available via silver-salt metathesis. A platinum(II) and a palladium(II) complex were isolated and a crystal structure of the former was examined. For the Pt(II) complex, luminescence is observed in solid state as well as in solution.

## 1. Introduction

The use of *N*-heterocyclic carbenes (NHC) has gained significant attention in recent years [1,2,3]. This ligand class has developed tremendously since its discovery by Arduengo et al. in 1991 [4]. This is due to their electronic properties, most notably the strong σ-donating character, which makes them useful ligands for the coordination of transition-metal ions [5,6]. Since then, complexes with NHC-type ligands have been intensively studied in the context of photophysical applications (e.g., with iron [7,8] or ruthenium [9]) as well as catalysis (e.g., with nickel [10], iron [11] or palladium [12]). Thereby, the ligand design has a strong influence on the properties of the resulting complexes, with steric and electronic properties having the most significant contributions [3,13,14]. The number of carbene sites, that surround the metal center, has been identified as one further relevant factor (i.e., the more carbene moieties, the stronger expressed are the observed effects) [15]. It has also been shown that the NHC scaffold can be varied; in this respect, imidazole- and benzimidazole-type systems represent the most prominent examples to be found in the available literature [6,16,17]. Possessing an imidazole-type substructure, caffeine has already experienced some use as an NHC ligand but the overall number of publications referring to this is still minor [14,15,16,17,18,19,20,21,22,23,24,25,26,27,28,29,30,31,32,33,34,35,36]. For example, the antimicrobial and anticancer properties of caffeine-derived NHC complexes have been studied thoroughly [17,20,25,30,31,32,33]. It has to emphasized that caffeine represents a promising starting point for the design of NHC-type ligands, due to its availability, low price and, most notably, green footprint compared to others. In here, the implementation of caffeine into a tridentate ligand framework, with pyridine as the central unit, is presented (Figure 1).

This particular strategy has already been used inter alia to prepare tridentate ligands with imidazole, benzimidazole and triazole units [6,37,38,39,40,41,42]. Concerning Pt(II) complexes, the NHC-pincer approach, using CCC- or CNC-type ligands, has been intensively studied [43,44,45,46,47,48]. The methylene bridges in the presented ligand allow for a more relaxed and ideal octahedral or square-planar coordination geometry—a factor that has also been shown to have a significant impact [49]. Notably, the ligand design offers plenty of options to further build upon, since the 4-position of the pyridine unit can easily be modified, e.g., by halogen atoms and, thus, enabling subsequent C-C coupling reactions [42]. Moreover, substituents others than methyl could be introduced at the *N*-atoms of the purine-base moieties (e.g., long alkyl chains in order to mediate the solubility).

## 2. Results

The precursor (or protoligand) synthesis was accomplished via a straightforward two-step procedure (Figure 2), in which commercially available 2,6-bis(bromomethyl)-pyridine (**1**) is first added to deprotonated theophylline (**2**). This reaction can be performed in neat DMF; however, the addition of water accelerates the deprotonation of the purine base. The overall reaction volume must be sufficiently large as the product swells enormously and turns the reaction mixture into a gel. Mechanical stirring is even required after some time has passed. The high viscosity of the reaction mixture also calls for an excess of theophylline to allow the reaction to proceed efficiently; the educt is later removed by copious washing to give **3** as a fluffy, white solid. Methylation of the imidazole unit is the next step. Mild methylating agents, such as Meerwein’s salt, should be used at an ambient temperature to avoid over-methylation. Compound **3** is barely soluble in CH_2_Cl_2_ but, nevertheless, the best results were obtained when the methylation was performed in this solvent and long reaction times were applied (i.e., up to 72 h, but at least 24 h). The yield of the two reaction steps is, owing to the solubility of the compounds, overall acceptable.

NHCs can be prepared following different procedures. Today, the most commonly used methods rely on the in situ generation of the carbene by a strong base, followed by the addition of a suitable transition-metal salt to form the complex [13,50]. Recently, a slightly varied approach has been developed which uses a weaker base or a metal salt, that already bears a ligand that can act as a base itself. Firstly, the base route was tested. Specifically, the deprotonation step, which involved the addition of KO*t*Bu or *n*-BuLi to a suspension of **4** in THF at –80 °C yielded a yellow solution. However, the addition of transition-metal salts to this solution—either directly or dissolved in a solution—did not afford the desired complexes. Notably, the reaction with Pd(OAc)_2_, which intrinsically carries its “own” base (i.e., the AcO^−^ counterion), also did not give the desired complex. Subsequently, the second common route was tested. This route involves the formation of a Ag(I) precursor complex which can then be used for salt metathesis.

The silver complex **5** was generated by heating the protoligand **4** in a mixture of CH_3_CN/CH_2_Cl_2_ with Ag_2_O at 50 °C for at least two hours and confirming the absence of the imidazolium C-H bond via ^1^H-NMR spectroscopy (Appendix A Appendix A). The other protons experienced a shift of their signals in the NMR spectrum: The pyridine protons were shifted upfield by 0.1 ppm (C-H triplet in 4-position) and 0.4 ppm (C-H doublet in 3-position), respectively. The signal of the methylene groups was upfield shifted by 0.15 ppm; whereas, the signals of the CH_3_-groups were all shifted downfield with the imidazolium N-CH_3_ featuring the largest shift of 0.08 ppm (the other two CH_3_-groups were only shifted marginally by 0.03 and 0.01 ppm, respectively). The ^13^C-NMR reveals a large shift of the signal assigned to the Ag(I)-bound carbon when compared to **4** (i.e., 189 pm vs. 140 ppm; Appendix A). This chemical shift is indicative of the carbene structure, and, according to Huyn, the value ranks it as strong δ–donor [51].

Basically, the Ag(I) complexes can be either monomeric species, as depicted in Figure 3, or form dimeric structures. The particular structure of these complexes is mainly dependent on the stoichiometry of the reaction (i.e., the number of Ag_2_O equivalents) as well as the used solvent. It has been reported that the solubility of Ag(I) salts in MeOH is lower when compared to CH_3_CN or CH_2_Cl_2_; thus, the formation of monomers is favored in these solvents [52]. The structure of the complex, in turn, represents an important factor for the subsequent salt-metathesis reaction with respect to the required stoichiometry. In the present case, ESI-MS gave strong evidence for the monomeric structure of **5** (Appendix A) Tandem MS, specifically travelling-wave ion-mobility MS (TWIM MS) coupled to gradient tandem MS (gMS^2^) is highly suited for the deconvolution and characterization of superimposed ions, which were generated via ESI. This particular method enables one to distinguish between organometallic ions of identical *m/z* values by, in simple words, net-charge and shape [53]. However, this technique is demanding and its application to the present system seems disproportionate.

Complex **5** was subsequently allowed to react with different transition-metal salts in CH_3_CN. The metals that were tested during our studies were iron(II), nickel(II), palladium(II), platinum(II) and ruthenium(III) (the results obtained for iron, nickel and ruthenium are summarized in the Appendix B). The Pd(II) and Pt(II) complexes were prepared from PdCl_2_ and K_2_PtCl_4_, respectively. In both cases, a yellow solution was obtained after the reaction with silver precursor **5** (Figure 3). Both complexes, **6** and **7**, were isolated after column chromatography using CH_3_CN/H_2_O/KNO_3_ (40:4:1 ratio) as the eluent, and SiO_2_ as the stationary phase, followed by precipitation with NH_4_PF_6_.

The ^1^H-NMR spectra of **6** and **7** revealed the absence of the imidazolium proton as well as a splitting of the signal of the methylene groups, suggesting a diastereotopic environment of those protons (see Figure 1, Appendix A).

Compared to Ag(I) complex **5** and protoligand **4**, the signals of the pyridine’s protons revealed a significant downfield shift—the shift of 0.3 ppm and 0.6 ppm for the triplet and doublet, respectively, was almost identical for both complexes (Appendix A). For the other signals, the Pd(II) complex **6** showed a slightly larger shift compared to its Pt(II) counterpart **7**. Again, the signal of the CH_2_ groups split into two doublets at 6.57 and 5.70 ppm for **6** and 6.40 and 5.42 ppm for **7**. This was already observed for similar complexes that also feature a methylene bridge between the carbene and the central binding unit. Signal splitting is attributed to the helical twisted structure which can form due to the ligand’s flexibility. This feature is absent when the carbene unit is directly attached to the pyridine ring [47,54,55,56,57,58,59,60]. The overall splitting of the signals, however, is larger for **7** compared to **6** (1.01 ppm vs. 0.87 ppm). The methyl groups for both complexes are similar again, with the Pd(II) complex’ imidazolium N-CH_3_ group revealing a 0.05 ppm downfield shift compared to **7**, with the other CH_3_ groups having similar shifts. The diastereotopic protons were further studied by variable temperature coalescence experiments to identify the coalescence temperature at which the conformational change becomes faster than the NMR timescale. Temperature-dependent experiments were conducted up to 100 °C. For complex **6**, at this temperature the signals just start to broaden; whereas, no change is observed for **7** (Appendix A). Thus, the coalescence temperature of the complexes is relatively high, probably around 120 °C for **6** and 140 °C or higher for **7**. This difference, when compared to other published systems [61], could be explained by the bulkiness of the caffeine—similarly high conformational rigidity has also been observed for other large systems [60].

The shift of the pyridine protons into the downfield region is due to the deshielding of the protons. Compared to **5** and **4**, the electron density in the pyridine ring is reduced due to the N-M bond which is now present. This effect is stronger for the proton in 4-position compared to those in 3/5-position due to the mesomeric connectivity with the nitrogen.

The ^13^C-NMR spectrum of **6** reveals the carbene shift to be at 174 ppm; (Appendix A) whereas for **7**, the proton shift is 171 ppm. This means that in either case the ligand is only moderately σ–donating when compared to other systems [51]. High-resolution ESI-MS measurements confirmed the identity of the complexes and excluded the presence of bridged species of higher nuclearity.

### 2.1. Crystallographic Data

Single crystals, suitable for X-ray structural analysis, were grown by slow diffusion of diethyl ether into a concentrated CH_3_CN solution of **7**, affording crystalline **7**·CH_3_CN (Figure 2).

The coordination of the platinum atom is square-planar with the two NHC groups in a *trans*-arrangement (C1-Pt1-C2 173.46(8)°). The remaining two coordination sites are occupied by the central pyridine moiety and a chlorido ligand, being oriented linearly to each other (N1-Pt1-Cl1 178.50(5)°). The two NHC planes are twisted out of the metal’s coordination plane by 39.06(6)° and 46.97(6)°, respectively, and are oriented almost perpendicular to each other (angle between C_3_N_2_ planes 84.04(8)°). The PF_6_^−^ counterion as well as the CH_3_CN molecules, which were present in the crystal structure, do not display any unusually close intermolecular contacts. The Pt-C bond lengths are virtually identical at 2.024(2) and 2.014(2) Å, which is in the typical range observed for Pt(II)-NHC complexes deposited in the Cambridge Structural Database (CSD) [62]. The observed values are very similar to what was published by Limbach et al. in 2011, who studied platinum complexes with CH_2_-bridged, tridendate imidazole NHC ligands [54]. However, Santra et al. could not provide a crystal structure for their benzimidazole derivative; these authors performed B3LYP DFT calculations instead, and the estimated bond lengths are also similar, though a bit larger than for the imidazole ones [61]. The C_carbene_-N bond lengths are in a range of 1.339(3) to 1.374(2) Å. These values are larger compared to those reported for the imidazole and benzimidazole derivatives, which is most likely due to the π back-bonding from the metal atom to the C-N π* orbital, which is stronger in this caffeine-derived system [63,64]. The crystallographic data are compiled in Appendix A.

### 2.2. Photophysical Properties

Platinum(II) complexes sometimes feature room-temperature luminescence in the solid state [65,66]. This behavior can be related to metallophilic Pt···Pt interactions and/or π-π interactions within the crystal lattice; it has been shown that modifications of the ligand scaffold can tune the emission [65,66,67,68]. Therefore, the photophysical properties of compounds **4**, **6** and **7** were studied.

There are a few notable aspects to be mentioned here. Firstly, while the solid complex is emissive when excited at 365 nm, the solutions only emit when the energy-rich 254 nm wavelength is used. This behavior can be related to the absorption spectra (Figure 3). The precursor shows strong absorption at 258 and 270 nm; whereas, an additional band at ca. 317 is present for the complexes (Table 1). Secondly, solvatochromism is absent—at least qualitatively and at room temperature.

The solid-state emission of the complex is unlikely to stem from Pt···Pt interactions within the lattice, as the closest interatomic distance in the aforementioned crystal structure is ca. 7.2 Å, and thus much larger than the 3–4 Å that are usually observed in luminescent Pt-complexes [68,69]. A similar argument can be made for π-π interactions, as the distances are quite large between the two pyridine units. However, there seems to be an inverted stacking, i.e., the 5-membered ring is located on top of the caffeine’s 6-membered ring with distances of 3.6–3.7 Å. The distance between caffeine’s carbonyl moieties (O3/O1) and the pyridine nitrogen atom is approximately 3 Å.

The emission spectra of **4**, **6** and **7** are very similar (Figure 4). The spectral features and the data derived from the crystal-structure analysis let us conclude that the observed emission of the complexes stems from the caffeine itself rather than from the metal centers. The red-shift of the emission of **6** relative to that of **7** is in good agreement with related series of complexes [70]. The maximum of the emission band is red-shifted and the overall absorption is broadened.

It was observed that the emission’s wavelength of **4** depended on the applied excitation wavelength (Figure 5a). This so called “red-edge effect” was first discovered for quinine, as the common fluorescence standard, and has, since then, been observed for many different compounds [71]. The underlying mechanism, however, is not so clear and has been explained by different hypotheses [71,72,73,74]. However, pinpointing to one explanation is not so easy for this compound. This is due to the strong H-bonding and dipole interactions the caffeine moiety might be involved in. These interactions are apparent from concentration dependent NMR shifts and are related to the strong dipole interactions with the solvent due to the dipole moment of the caffeine as well as the free rotation around the methyl linkage to the pyridine.

This same dependence has also been observed for the **6** and **7** (Figure 5b,c), although in a much less pronounced fashion. This provides us insight into the mechanism of this phenomenon. The aforementioned temperature-dependent NMR studies already indicated the high importance of the methyl bridge. A similar effect is observed here. There is a trend from protoligand **4** to **6** to **7** regarding the red-shift of the emission; this trend qualitatively matches the coalescence behavior. Whereas, in **4** all groups may rotate freely, this is less so for the complexes due to the metal-carbon bond. While **6** still changes its conformation relatively fast, complex **7** is more rigid and less fluctuating. This behavior is expressed by the position of the emission band at different excitation wavelengths: For protologand **4**, the observed Δnm is 79; whereas, lower values were found for **6** and **7** (Δnm of 15.5 and 2.5 for **7**, respectively; Table 1).

The solid-state emission of the compounds was evaluated via a plate reader; however, only **4** was emissive enough to collect data (Figure 6). Overall the weak solid-state fluorescence of the complexes, in particular of **7**, is most likely due to the quenching effects. The substitution of chloride by alkyne derivatives or heavy-atom ligands (e.g., AsPh_3_ or SbPh_3_) is expected to improve the photophysical performance significantly [62,69,75].

These results are interesting when contrasted with the literature known CCC- and CNC-type NHC pincer complexes **8** and **9** (Figure 4) [47,48].

While it lacks the methyl bridge and the coordinating nitrogen, complex **8** is highly emissive in solution as well as in the solid state. It features absorption bands at 265, 323, 416 and 441 nm, the first two of which are comparable to the complexes presented here. They have been attributed to mixed metal to ligand charge transfer and ligand-centred (MLCT-LC) transitions [76]. It emits at 449 and 474 nm, which is comparable to 441 nm of **7**. Complex **9** is a CNC-type complex which very closely resembles **7**, only missing the methyl bridge and having imidazole units instead of caffeine. Unlike **7**, **9** is highly emissive and features green to orange fluorescence, depending on the water content. The authors attributed this behaviour and the emission to the Pt-Pt dimers due to the proximity (3.5 Å).

## 3. Discussion and Conclusions

The Pd(II), Pt(II) and Ag(I) complexes of a new tridentate ligand, which contained two caffeine-derived NHC moieties, were synthesized. The overall straightforward approach is highly promising since the ligand scaffold can readily be modified via the pyridine- or the purine-based sub-units. It remains to be tested whether the platinum or the silver complex exhibit anti-tumor or anti-microbial activity [25,32]. Moreover, the Pd(II) complex is currently evaluated regarding its catalytic activity in, e.g., cross-coupling reactions [18]. The emission and absorption properties of **5**, **6**, and **7** are reported and a red-edge excitation shift has been observed. The synthesis and photophysical characterization of derivatives of Pt(II) complex **7** (with alkyne or heavy-atom ligands) is also the subject of ongoing research.

## 4. Materials and Methods

### General Procedures

Unless stated otherwise, all reactions were carried out using standard Schlenk techniques in a dry nitrogen atmosphere. Dry solvents were purchased from Sigma-Aldrich (Darmstadt, Germany) and used as received. Theophylline and 2,6-bis(bromomethyl)pyridine were purchased from TCI (Eschborn, Germany), anhydrous FeBr_2_ was purchased from abcr (Karlsruhe, Germany), the other transition-metal salts were purchased from TCI or Sigma-Aldrich. Glassware was oven-dried at 110 °C prior to use. NMR spectra were recorded on a AVANCEI300 MHz, AVANCE II 400 MHz or AVANCE III 600 MHz spectrometer (all from Bruker BioSpin MRI GmBH, Ettlingen, Germany) with a cryo-probehead in deuterated solvents (euriso-top, Saarbrücken, Germany) at 25 °C. Standard parameters for processing of ^1^H-NMR spectra were line broadening of 0.3 and of ^13^C-NMR spectra with line broadening of 1. Chemical shifts are reported in ppm and are referenced using the residual solvent signal. High-resolution electrospray ionization time-of-flight mass spectrometry (ESI-TOF MS) was performed on an ESI-(Q)-TOF-MS MICROTOF II (Bruker Daltonics GmBH & Co KG, Karlsruhe, Germany) mass spectrometer. The single-crystal X-ray intensity data for 7·CH_3_CN were collected on a Bruker-Nonius Kappa-CCD diffractometer (Bruker ACS Advanced X-Ray Solutions, Karlsruhe, Germany) equipped with a Mo-K_α_ IµS microfocus source and an Apex2 CCD detector, at *T* = 120(2) K. The crystal structure was solved with SHELXT-2018/3 [77] and refined by full matrix least-squares methods on *F*^2^ with SHELXL-2018/3, [77] using the Olex 1.2 environment [78]. Multi-scan absorption correction was applied to the intensity data [26]. CCDC 2157149 contains the supplementary crystallographic data for this paper. These data can be obtained free of charge from The Cambridge Crystallographic Data Centre (CCDC; http://www.ccdc.cam.ac.uk (accessed on 20 May 2022). Fluorescence spectra were recorded on a JASCO FP-8300 spectrometer equipped with a Peltier element.

*Synthesis of 7,7′-(pyridine-2,6-diylbis(methylene))bis(1,3-dimethyl-xanthine)* (**3**). Theophylline (2.0 g, 11.1 mmol) was dissolved in DMF (80 mL). K_2_CO_3_ (3.0 g, 21.71 mmol) in H_2_O (10 mL) was added to the solution and the resulting suspension was stirred for 3 h at ambient conditions. 2,6-Bis(bromomethyl)pyridine (500 mg, 1.89 mmol) in DMF (5 mL) was added and the reaction mixture was stirred overnight while manually breaking up the gel when the mixture became too viscous. Water (50 mL) was added, the solid was filtered off and washed with aq. NaOH (1 M, 40 mL), H_2_O (400 mL), EtOH (200 mL) and subsequently dried in vacuum to give the product as a colourless, cotton-like solid (650 mg, 1.40 mmol, 74%). ^1^H-NMR (CDCl_3_, 300 MHz): *δ* = 7.70 (t, 1H, *J* = 7.7 Hz, a-H), 7.66 (s, 2H, d-H), 7.33 (d, 2H, *J* = 7.7 Hz, b-H), 5.52 (s, 4H, c-H), 3.61 (s, 6H, 1′-H), 3.35 (s, 6H, 3′-H) ppm. ^13^C-NMR (CDCl_3_, 75 MHz): *δ* = 154.7 (1-C), 151.7 (4/5-C), 151.2 (5/4-C), 148.8 (d-C), 141.9 (3-C), 138.5 (a-C), 121.9 (b-C), 106.6 (2-C), 50.9 (c-C), 29.8 (3′-C), 27.9 (1′-C) ppm. HRMS (ESI-TOF, *m*/*z*): 486.1614, [C_21_H_21_N_9_O_4_ + Na]^+^ (M + Na^+^) requires 486.1609.

*Synthesis of 7,7′-(pyridine-2,6-diylbis(methylene))bis(1,3,9-trimethyl-xanthinium)bis(tetrafluoborate)* (**4**). Compound **3** (500 mg, 1.08 mmol) was suspended in anhydrous CH_2_Cl_2_ (40 mL); Me_3_OBF_4_ (638 mg, 4.32 mmol) and K_2_CO_3_ (100 mg) were added to the reaction mixture and stirred overnight. The solvent was removed under reduced pressure, subsequently MeOH (50 mL) was added and removed under reduced pressure. This sequence was repeated three times. The resulting solid was dissolved in a minimal amount of CH_3_CN, filtered and then precipitated by the addition of diethyl ether. The colourless solid was dried to yield the product (600 mg, 0.9 mmol, 84%). ^1^H-NMR (CD_3_CN, 300 MHz): *δ* = 8.76 (s, 2 H, d-H), 7.89 (t, 1H, *J* = 7.7 Hz, a-H), 7.52 (d, 2H, *J* = 7.7 Hz, b-H), 5.70 (s, 4H, c-H), 4.15 (s, 6H, 9′-H), 3.74 (s, 6H, 1′-H), 3.25 (s, 6H, 3′-H) ppm. ^13^C-NMR (CD_3_CN, 75 MHz): *δ* = 154.6 (1-C), 153.4 (4/5-C), 151.5 (5/4-C), 140.9 (d-C), 140.3 (a-C), 140.2 (3-C), 124.4 (b-C), 108.7 (2-C), 53.4 (c-C), 38.3 (9′-C), 32.3 (3′-C), 29.2 (1′-C) ppm. HRMS (ESI-TOF, *m*/*z*): 246.6088, [C_23_H_27_N_9_O_4_]^2+^ (M^2+^) requires 246.6088.

*Synthesis of the Ag(I) complex 7,7′-(pyridine-2,6-diylbis(methylene))-bis(1,3,9-trimethyl-xanthinyliden) silver tetrafluoroborate* (**5**). Protoligand **4** (2.0 g, 3.0 mmol) was dissolved in a mixture of CH_3_CN (50 mL) and CH_2_Cl_2_ (20 mL) as well as Ag_2_O (2.0 g, 8.63 mmol) were added. The mixture was stirred at 50 °C for 2 h. The reaction progress was monitored by ^1^H-NMR spectroscopy and heating was stopped when the signal of the imidazolium proton had disappeared. The reaction mixture was filtered to remove the suspended solid and then precipitated into diethyl ether to yield the product as a grey solid (1.90 g. 1.38 mmol, 46%). ^1^H-NMR (CD_3_CN, 300 MHz): *δ* = 7.73 (t, 1H, *J* = 7.7 Hz, a-H), 7.10 (d, 2H, *J* = 7.7 Hz, b-H), 5.50 (s, 4H, c-H), 4.25 (s, 6H, 9′-H), 3.76 (s, 6H, 1′-H), 3.24 (s, 6H, 3′-H) ppm. ^13^C-NMR (CD_3_CN, 75 MHz): *δ* = 189.7 (d-C), 156.4 (1-C), 154.9 (4/5-C), 151.9 (5/4-C), 142.0 (a-C), 140.3 (3-C), 122.9 (b-C), 110.2 (2-C), 54.7 (c-C), 40.5 (9′-C), 32.4 (3′-C), 29.0 (1′-C) ppm. HRMS (ESI-TOF, *m*/*z*): 598.1055, [C_23_H_25_AgN_9_O_4_]^+^ (M^+^) requires 598.1075.

*Synthesis of the Pd(II) complex 7,7′-(pyridine-2,6-diylbis(methylene))-bis(1,3,9-trimethyl-xanthinyliden) palladiumchlorido hexafluorophosphate* (**6**)**.** PdCl_2_ (241 mg, 1.36 mmol) was dissolved in anhydrous CH_3_CN (20 mL) at 60 °C and after 1 h the silver salt 5 (1.0 g, 1.36 mmol) was added. The resulting yellow-orange suspension was filtered after 18 h and the solvent was removed under reduced pressure. The residue was purified by column chromatography (SiO_2_, 40:4:1, CH_3_CN, H_2_O, KNO_3_, sat.) and the second yellow fraction was collected. NH_4_PF_6_ (1.0 g) in H_2_O (10 mL) was added and the solution was extracted with CH_2_Cl_2_ (50 mL). The solvent was removed under reduced pressure to yield the product as a yellow solid (280 mg, 0.44 mmol, 33%). ^1^H-NMR (CD_3_CN, 300 MHz): *δ* = 8.07 (t, 1H, *J* = 7.7 Hz, a-H), 7.73 (d, 2H, *J* = 7.70 Hz, b-H), 6.43 (d, 2H, *J* = 15.2 Hz, c-H), 5.70 (d, 2H, *J* = 15.2 Hz, c-H), 4.28 (s, 6H, 9′-H), 3.73 (s, 6H, 1′-H), 3.32 (s, 6H, 3′-H) ppm. ^13^C-NMR (CD_3_CN, 75 MHz): *δ* = 174.2 (d-C), 155.6 (1-C), 154.8 (4/5-C), 151.9 (5/4-C), 143.1 (a-C), 141.9 (3-C), 127.1 (b-C), 109.4 (2-C), 53.7 (c-C), 39.4 (9′-C), 32.8 (3′-C), 28.9 (1′-C) ppm. HRMS (ESI-TOF, *m*/*z*): 632.0730, [C_23_H_25_ClN_9_O_4_Pd]^+^ (M^+^) requires 632.0730.

*Synthesis of Pt(II) complex 7,7′-(pyridine-2,6-diylbis(methylene))-bis(1,3,9-trimethyl-xanthinyliden) platninumchlorido hexafluorophosphate* (**7**)**.** The silver salt **5** (1.0 g, 1.36 mmol) and K_2_PtCl_4_ (564 g, 1.36 mmol) were dissolved in anhydrous CH_3_CN (20 mL) and the reaction mixture was stirred overnight at 60 °C. The resulting light-yellow suspension was filtered and the solvent was removed under reduced pressure. The residue was purified by column chromatography (SiO_2_, 40:4:1, CH_3_CN, H_2_O, KNO_3_, sat.) and the second yellow fraction was collected. NH_4_PF_6_ (1.0 g) in H_2_O (10 mL) was added and the solution extracted with CH_2_Cl_2_ (50 mL). The solvent was removed under reduced pressure to yield the product as a slightly yellow solid (140 mg, 0.19 mmol, 14%). ^1^H-NMR (CD_3_CN, 300 MHz): *δ* = 8.08 (t, 1H, *J* = 7.7 Hz, a-H), 7.75 (d, 2H, *J* = 7.70 Hz, b-H), 6.57 (d, 2H, *J* = 15.2 Hz, c-H), 5.41 (d, 2H, *J* = 15.2 Hz, c-H), 4.24 (s, 6H, 9′-H), 3.74 (s, 6H, 1′-H), 3.32 (s, 6H, 3′-H) ppm. ^13^C-NMR (CD_3_CN, 75 MHz): *δ* = 171.2 (d-C), 155.8 (1-C), 155.1 (4/5-C), 152.2 (5/4-C), 142.9 a-C), 142.6 3-C), 127.7 (b-C), 109.4 (2-C), 53.9 (c-C), 33.1 (9**′**-C), 30.1 (3′-C), 29.2 (1′-C) ppm. HRMS (ESI-TOF, *m*/*z*): 721.1334, [C_23_H_25_ClN_9_O_4_Pt]^+^ (M^+^) requires 721.1360.

## Data Availability

The data for the synthesis and structural characterization of all compounds are stored at the FSU Jena.

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
