# Peer review of "Noble Metal Complexes of a Bis-Caffeine Containing NHC Ligand"

_molecules, 2022, doi:10.3390/molecules27134316_

Round 1

Reviewer 1 Report

The introduction should be expanded, in particular on the use of caffeine as NHC ligand for metal complexes: the authors report references 18-23, but they are quite limited. The number of papers on this topic remains still limited, so it is important to include all the connected citations.

The synthesis and NMR characterization of the complexes is well done and needs minor improvements. For example:

1) comment on the difference observed for the CH2 signal in the NMR spectra of silver and Pd/Pt complexes.

2) is the coupling between carbene carbon and silver or platinum visible in in the 13C NMR spectra?

3) line 136: carbon shift

4) line 137: sigma (greek letter) not delta must be used for the ligand donation.

5) pay attention that 4 is not the ligand but it is its precursor. The ligand can be obtained only by deprotonation.

The main problem of the manuscript is the quality of the photophysical characterization.

1) why the absorption of 4 in Figure 4a is constant and also scattering might be present in the range 300-400 nm?

2) the “ligand” has no absorption at 317 nm (apart from that described in the previous point 1) so that the origin of the emission when the sample is excited at 317 nm is not clear and need to be justified.

3) Fig5b. the emission of the “ligand” in the two experiments is shifted of ca. 150 nm (a table should be added with the feature of the compounds). This large shift with the excitation wavelength must be justified.

4) Figure 4b (as well as lines 179-184) should be removed since it is difficult to see the emission.

5) lines 204-205: what is the meaning of “fluorescence …due to chlorido ligand”? in order to understand the origin of emission other studies should be made (for example determination of lifetimes). This will be fundamental also to understand if the emission is fluorescence or phosphorescence.

Author Response

Reviewer #1:

Comments and Suggestions for Authors

The introduction should be expanded, in particular on the use of caffeine as NHC ligand for metal complexes: the authors report references 18-23, but they are quite limited. The number of papers on this topic remains still limited, so it is important to include all the connected citations.

We have included references 24-45 in the introduction.

The synthesis and NMR characterization of the complexes is well done and needs minor improvements. For example:

  • comment on the difference observed for the CH2 signal in the NMR spectra of silver and Pd/Pt complexes.

We thank the reviewer for this comment and have added the following passages to the main text:
“This was already observed for similar complexes that also feature a methylene bridge between the carbene and the central binding unit. It is attributed to the helical twisted structure which can form due to the flexibility. It is absent when the carbene unit is directly attached to the pyridine. [54,61-67]” and “The diastereotopic protons were further studied by variable temperature coalescence experiments to find the coalescence temperature at which the conformational change becomes faster than the NMR timescale. Temperature experiments were conducted up to 100 °C. For complex 6, at this temperature the signals just start to broaden; whereas, no change is observed for 7. In both cases this means the coalescence temperature is relatively high, probably around 120 °C for 6 and 140 °C or higher for 7. This difference, when compared to other published systems [62] could be explained by the bulkiness of the caffeine as this high conformational rigidity has been observed for similarly large systems [67].”

In addition, the collected spectra were added to the supporting information.

2) is the coupling between carbene carbon and silver or platinum visible in in the 13C NMR spectra?

            No, there is no coupling visible for any of the metal complexes in the 13C NMR spectra.

3) line 136: carbon shift

            We thank the reviewer for pointing this out

4) line 137: sigma (greek letter) not delta must be used for the ligand donation.

            We thank the reviewer for pointing this out

5) pay attention that 4 is not the ligand but it is its precursor. The ligand can be obtained only by deprotonation.

We thank the reviewer for pointing this out and have changed it in the manuscript accordingly

The main problem of the manuscript is the quality of the photophysical characterization.

  • why the absorption of 4 in Figure 4a is constant and also scattering might be present in the range 300-400 nm?

We thank the reviewer for pointing this out. We have repeated the absorption measurement for the precursor at a more suitable concentration. When we performed the experiment, the concentration was apparently too low which resulted in the artifact (constant absorption) upon normalization.

We do not argue for scattering as the UVvis spectrometer uses a dual beam setup.

  • the “ligand” has no absorption at 317 nm (apart from that described in the previous point 1) so that the origin of the emission when the sample is excited at 317 nm is not clear and need to be justified.

We thank the reviewer for this comment. When repeating the experiment with higher concentration, the absorption after 300 nm became more apparent. There is only very weak absorption at 317 nm, but it does indeed absorb.

Figure 1: Absorption of precursor 4 with extreme concentration

  • the emission of the “ligand” in the two experiments is shifted of ca. 150 nm (a table should be added with the feature of the compounds). This large shift with the excitation wavelength must be justified.

We thank the reviewer for this comment. We checked for excitation wavelength dependence and have added the following passage to the main text:

“One observation during the measurements was the strong dependence of the wavelength emission of 4 on the corresponding excitation wavelength (Figure 6a). This so called “red-edge effect” was first discovered on the common fluorescence standard quinine and has, since then, been observed for many different compounds [78]. The underlying mechanism, however, is not so clear and different hypotheses exist [78-81]. Pinpointing to one explanation is not so easy for this compound. This is due to the strong H-bonding and dipole interactions that can form from the caffeine unit which are apparent from concentration dependent NMR shifts; strong dipole interactions with the solvent due to the dipole moment of the caffeine; free rotation around the methyl linkage to the pyridine.

(a)

(b)

(c)

Figure 6.  Room-temperature emission spectra of 4 (a), 6 (b) and 7 (c) in solution after excitation at varying wavelengths ( 10-5M in CH3CN) .

Table 1. Collection of spectroscopic data.

Compound

Absorption / nm

Emission / nm

4

267

321.5a, 400.5b

6

268, 316

435.5a, 420b

7

269, 321

441a, 443.5b

a Excited at 267 nm, b excited at 337 nm

This same dependence has also been observed for the Pd and Pt complex (Figure 6b and 6c), although in much less pronounced fashion. This provides us insight into the mechanism of this phenomenon. From the temperature dependent NMR studies, we already showed the high importance of the methyl bridge and here it is not different. There is a trend from the precursor to the Pd complex to the Pt complex with the red-shift and it also matches the coalescence. Whereas, in 4 all groups may rotate freely, this is less so for the complexes due to the metal-carbon bond. While the Pd complex changes conformation faster, the Pt complex is more rigid and this also matches with the observed shift in emission. This explains why for 4 the observed Δnm is 79 while for 6 Δnm is 15.5 and only 2.5 for 7 (Table 1).“

4) Figure 4b (as well as lines 179-184) should be removed since it is difficult to see the emission.

We thank the reviewer for this comment and have removed Figure 4b and altered the text above to read
So, the photophysical properties of compounds 4, 6 and 7 were studied.”

5) lines 204-205: what is the meaning of “fluorescence …due to chlorido ligand”? in order to understand the origin of emission other studies should be made (for example determination of lifetimes). This will be fundamental also to understand if the emission is fluorescence or phosphorescence.

            We wholeheartedly agree with the reviewer here. Unraveling the photophysics of this complex is of interest to us. This is why we are cooperating with the group of Prof. Dr. Axel Klein from cologne to study the compound from this publication as well as a few additional complexes.

We have checked for phosphorescence of the compounds but our setup only allows us to measure phosphorescence with a lifetime of at least 1 ms. With these constraints we could not measure any long-lived phosphorescence. In addition, to properly elucidate the photophysics, more sophisticated experiments such as measuring at 77K should also be done for which we lack the experimental setup. This would also allow for proper determination of lifetimes as they are expected to be quite short. In addition: The presented results prompt the question if proper red-edge excitation spectroscopy should be performed. The elucidation of the fluorescent microstates warrants further investigation by photophysical specialists.

Reviewer 2 Report

The research presented by Schubert and coworkers describes the preparation of a bis-NHC compound (4) and it use as a ligand for several noble metals (Ag, Pd, Pt). The characterization of these metal complexes has been performed by NMR in these three cases. X-ray diffraction has also been described for one of this compounds (7). Some other photophysical properties have been included for these compounds and attempts to synthesize other metal complexes (Ru, Fe,…) have been described in the last part.

This research has been made with care and all discussion and conclusion are in accordance with detailed experimental observations. Experimental procedures and characterizations are described carefully and the introductory part and references are adequate for the research described. In general, the paper is easy to read and authors have made an effort to explain all data extracted from the research. In my opinion, although the presented seems to be quite scarce (single ligand for three metal complexes and their characterizations), it fulfills all requirements for being published in Molecules.

Author Response

Reviewer #2:

Comments and Suggestions for Authors

The research presented by Schubert and coworkers describes the preparation of a bis-NHC compound (4) and it use as a ligand for several noble metals (Ag, Pd, Pt). The characterization of these metal complexes has been performed by NMR in these three cases. X-ray diffraction has also been described for one of this compounds (7). Some other photophysical properties have been included for these compounds and attempts to synthesize other metal complexes (Ru, Fe,…) have been described in the last part.

This research has been made with care and all discussion and conclusion are in accordance with detailed experimental observations. Experimental procedures and characterizations are described carefully and the introductory part and references are adequate for the research described. In general, the paper is easy to read and authors have made an effort to explain all data extracted from the research. In my opinion, although the presented seems to be quite scarce (single ligand for three metal complexes and their characterizations), it fulfills all requirements for being published in Molecules.

Reviewer 3 Report

The current manuscript by  Oliver Bysewski et al describes the synthesis, characterization of three metal (Ag, Pd and Pt) complexes with NHC tridentate pyridine-based ligand. The authors have observed luminescence for platinum complex in both solid state as well as solution. 

Many points should be reviewed by the authors: 

1-     The title of the manuscript does not underline the photophysical properties.

2-     The introduction should be enriched with examples of luminescent Pt complexes.

3-     In abstract ‘N-Hetrocyclic carbenes (NHC)’ should be corrected with (NHCs).

4-     The IUPAC nomenclature of NHC metal complexes should be indicated.

5-     Pag. 2 line 53 ‘This ligand design offers plenty of options to further build upon, 53 since the 4-position of the pyridine unit can easily be modified, e.g., by halogen atoms and, 54 thus, employed in C-C coupling reactions.’ A reference for this sentence should be added.

6-     In scheme 2 ‘including the atom numbering’ sentence is not correct. It would be better to write: ‘including the indication of the atoms’ or similar.

7-     In figure 2 there is a mistake ‘asterx’.

8-     The authors must justify the shift of the peaks at 7.11 and 7.73 ppm in complex 5 to lower fields for complexes 6 and 7.

9-     It would be appropriate to report in the supplementary part, for 1H and 13C, the structures of the compounds and the list of chemical shifts.

10- In scheme 3 it would be appropriate to report the protons 1 ', 3', etc, indicated in the experimental part.

11- All carbon atoms should be assigned.

12- Page 9 line 262. Why is methanol added and removed under reduced pressure?

13- The silver complex was simply filtered, what size filter paper was used? My experience is that Ag2O is not easily stop.

14- In figure 4 ligand should be corrected with protoligand.

15- The conclusions should be reviewed. No luminescence data is reported.

Author Response

Reviewer #3:

Comments and Suggestions for Authors

The current manuscript by Oliver Bysewski et al describes the synthesis, characterization of three metal (Ag, Pd and Pt) complexes with NHC tridentate pyridine-based ligand. The authors have observed luminescence for platinum complex in both solid state as well as solution. 

Many points should be reviewed by the authors: 

  • The title of the manuscript does not underline the photophysical properties.

The photophysical properties are not the main focus of the paper. There is no in-depth analysis, just basic characterization of the complexes.

  • The introduction should be enriched with examples of luminescent Pt complexes.

We have added the following passage to the introduction:
“For platinum complexes in particular, the NHC-pincer approach has been studied for a CCC and CNC-type complex.[50-55]”

In addition we have added to the results:

“These results are interesting when contrasted with the literature known CCC- and CNC-type NHC pincer complexes 8 and 9 (Scheme 4).[54,81]

Scheme 3. Schematic representation of 8 and 9

While it lacks the methyl bridge and the coordinating nitrogen, complex 8 is highly emissive in solution as well as in the solid state. It features absorption bands at 265, 323, 416 and 441 nm, the first two of which are comparable to the complexes presented here. They have been attributed to mixed metal to ligand charge transfer and ligand centered (MLCT-LC) transitions.[82] It’s emits at 449 and 474 nm which is comparable to 441 nm of 7.

Complex 9 is a CNC type complex which very closely resembles 7, only missing the methyl bridge and having imidazole units instead of caffeine. Unlike 7, 9 is highly emissive and features green to orange fluorescence, depending on the water content. The authors attributed this behaviour and the emission to the Pt-Pt dimers due to the proximity (3.5 Å).“

3-     In abstract ‘N-Hetrocyclic carbenes (NHC)’ should be corrected with (NHCs).

We thank the reviewer and have corrected this in the abstract.

4-     The IUPAC nomenclature of NHC metal complexes should be indicated.

We thank the reviewer and have added IUPAC conformant nomenclature to the complexes

5-     Pag. 2 line 53 ‘This ligand design offers plenty of options to further build upon, 53 since the 4-position of the pyridine unit can easily be modified, e.g., by halogen atoms and, 54 thus, employed in C-C coupling reactions.’ A reference for this sentence should be added.

                  We have added reference 49 here.

6-     In scheme 2 ‘including the atom numbering’ sentence is not correct. It would be better to write: ‘including the indication of the atoms’ or similar.

We have corrected this in the revised version.

7-     In figure 2 there is a mistake ‘asterx’.

We have corrected this in the revised version.

8-     The authors must justify the shift of the peaks at 7.11 and 7.73 ppm in complex 5 to lower fields for complexes 6 and 7.

We have added the following at lines 155-159:

“The shift of the pyridine protons into the downfield region can be explained by the deshielding of the protons. Compared to 5 and 4, the electron density is reduced in the pyridine due to the N-M bond which is now present. This is stronger for the proton in the para position compared to the meta due to the mesomeric connectivity with the nitrogen.”

9-     It would be appropriate to report in the supplementary part, for 1H and 13C, the structures of the compounds and the list of chemical shifts.

10- In scheme 3 it would be appropriate to report the protons 1 ', 3', etc, indicated in the experimental part.

                  We have assigned the protons in the corresponding Schemes 2 and 3

11- All carbon atoms should be assigned.

12- Page 9 line 262. Why is methanol added and removed under reduced pressure?

We thank the reviewer for pointing this procedure out and are happy to elaborate. The meerwein’s salt which is used for methylation of the xanthine is used in excess and thus there is still some left. Even if stoichiometric amounts are used, methanol should be added regardless during the workup to make sure that any unreacted methylating agent is quenched. The quenching product is dimethyl ether and this is easily removed under reduced pressure.

13- The silver complex was simply filtered, what size filter paper was used? My experience is that Ag2O is not easily stop.

We agree with the reviewer. Filter paper usually is not able to fully remove the silver which is why PTFE syringe filters (0.45 μm or 0.22 μm) of varying pore size are used instead. While more tedious, as these are very prone to clogging, the results are satisfactory.

14- In figure 4 ligand should be corrected with protoligand.

      This has been corrected.

15- The conclusions should be reviewed. No luminescence data is reported.

                  The following part has been added to the conclusion:

“The emission and absorption properties of 5, 6, and 7 are reported and a red-edge excitation shift has been observed.”

Round 2

Reviewer 1 Report

The authors have addressed all the comments of the reviewers and the quality of the manuscript has significantly improved.

For this reason I suggest its acceptance. I would like only to suggest an improvement in the quality of the figures: in figs. 4-6 it would be better to add the number of the complexes and proligand in the legend, so that the correspondence (for example in the table) will be easier for the reader.

Author Response

Dear Editor,
we are highly pleased to read that our Manuscript “Noble metal complexes of a bis-caffeine containing NHC ligand” will be accepted for publication after minor revision. We thank the referees for evaluating the manuscript and their positive feedback. All minor issues, as listed by the reviewers have been included (i.e., improvement of Figures 4-6). Furthermore, the manuscript was carefully revised regarding the language style. Please find enclosed the revised manuscript and all other required files. We looking forward to the official acceptance of our manuscript.

Please contact me for any further questions.

Best regards

Ulrich Schubert

Reviewer 3 Report

The authors replied in detail to all my comments, in this light, I recommend publication of this manuscript in its current form.

Author Response

(The authors gave the same response as above.)
